# Ethanol-Quenching Introduced Oxygen Vacancies in Strontium Titanate Surface and the Enhanced Photocatalytic Activity

**DOI:** 10.3390/nano9060883

**Published:** 2019-06-14

**Authors:** Yang Xiao, Shihao Chen, Yinhai Wang, Zhengfa Hu, Hui Zhao, Wei Xie

**Affiliations:** 1School of Physics & Optoelectronic Engineering, Guangdong University of Technology, Guangzhou 510006, China; xiaognay@163.com (Y.X.); shchenbhx@163.com (S.C.); kkhui@gdut.edu.cn (H.Z.); 2Synergy Innovation Institute for Modern Industries, Guangdong University of Technology, Dongyuan 517500, China; zhfhu@gdut.edu.cn; 3School of Physics Science and Technology, Lingnan Normal University, Zhanjiang 524048, China; xiewei@lingnan.edu.cn

**Keywords:** photocatslysis, SrTiO_3_, ethanol-quenching, oxygen vacancies, photodegradation, photocatlytic H_2_ generation

## Abstract

Modification of the surface properties of SrTiO_3_ crystals by regulating the reaction environment in order to improve the photocatalytic activity has been widely studied. However, the development of a facile, effective, and universal method to improve the photocatalytic activity of these crystals remains an enormous challenge. We have developed a simple method to modify the surface environment of SrTiO_3_ by ethanol quenching, which results in enhanced UV, visible and infrared light absorption and photocatalytic performance. The SrTiO_3_ nanocrystals were preheated to 800 °C and immediately quenched by submersion in ethanol. X-ray diffraction patterns, electron paramagnetic resonance spectra, and X-ray photoelectron spectra indicated that upon rapid ethanol quenching, the interaction between hot SrTiO_3_ and ethanol led to the introduction of a high concentration of oxygen vacancies on the surface of the SrTiO_3_ lattice. Consequently, to maintain the regional charge balance of SrTiO_3_, Sr^2+^ could be substituted for Ti^4+^. Moreover, oxygen vacancies induced localized states into the band gap of the modified SrTiO_3_ and acted as photoinduced charge traps, thus promoting the photocatalytic activity. The improved photocatalytic performance of the modified SrTiO_3_ was demonstrated by using it for the decomposition of rhodamine B and production of H_2_ from water under visible or solar light.

## 1. Introduction

Since the discovery of photoelectrochemical water splitting on a titania electrode by Fujishima and Honda in 1972, photocatalysis using semiconductors has been widely studied [1]. Semiconductor-based photocatalysts are capable of directly converting solar energy to chemical energy, which provides a facile approach for environmental protection and H_2_ production under sunlight irradiation [2,3,4]. Among these semiconductors, titanium dioxide (TiO_2_) has attracted special interest because of its chemical stability, non-toxicity, and low cost [5,6]. In addition, Strontium titanate (SrTiO_3_), a perovskite-type oxide, has been classified a wide-band gap (3.1–3.7 eV) semiconductor photocatalyst in the field of light energy exploitation. SrTiO_3_ possesses various outstanding physical and chemical properties such as good chemical/catalytic stability, suitable band position, and susceptibility to change by other substance. Notably, the conduction band of SrTiO_3_ is more negative than that of TiO_2_, which is beneficial for photocatalysis [7]. However, because of its large band-gap, SrTiO_3_ can be utilized only in the UV region of sunlight, which largely restricts its practical application in photocatalysis [8,9].

Many studies have attempted to expand the spectral response of SrTiO_3_ by utilizing methods such as doping with a metal/nonmetal, or co-doping with a nonmetal to form a new intermediate gap state between the valence band (VB) and the conduction band (CB) [10,11,12,13,14]. Moreover, mixing of the energy levels of the dopant and host elements can change the position of the VB or CB, resulting in increased visible light absorption [9]. However, the rapid rate of recombination of the electron-hole pairs in SrTiO_3_ results in poor photocatalytic efficiency. One of the methods to improve the separation efficiency of photogenerated electron-hole pairs involves combination with other noble metals (usually Pt, Au, and Rh) or semiconductors on the surface [15,16,17,18]. In the other hand, the important role played by surface oxygen vacancies in improving the photocatalytic performance is well known. Surface oxygen vacancies can be used as surface adsorption sites and photoinduced charge traps, where the charge shifts to the adsorbed compound, so that the recombination of photogenerated electron-holes is prevented and the photocatalytic performance is improved [19,20]. Therefore, introducing oxygen vacancies on the surface of the SrTiO_3_ lattice seems to be a feasible process to modify the surface environment of SrTiO_3_. This process could improve the adsorption capacity of SrTiO_3_ and inhibit the recombination of photogenerated electron-hole pairs, thus improving the photocatalytic performance of SrTiO_3_ [21].

In 2011, Mao and co-worker developed black TiO_2_ by introducing disorder in the surface layer of TiO_2_ via hydrogenation in a high H_2_-pressure atmosphere at about 200 °C for 5 days. This modification greatly enhanced the solar light harvesting efficiency and photocatalytic activity of TiO_2_ [22]. Subsequently, various methods were developed to synthesize hydrogenated or reduced semiconductors such as TiO_2_, ZnO and WO_3_ with surface oxygen defects for improving their photocatalytic performance [23,24,25,26]. Unsurprisingly, hydrogenated or reduced SrTiO_3_ was also synthesized. Tan et al. introduced oxygen vacancies on the SrTiO_3_ surface by heating it to 300–375 °C under Ar atmosphere, to control any solid-state reaction between NaBH_4_ and SrTiO_3_, which improved its photocatalytic activity for H_2_ generation under UV-vis irradiation [19]. Zhao et al. prepared black SrTiO_3_ with abundant Ti^3+^ cations and oxygen vacancies by reduction using molten aluminum at 500 °C. The black SrTiO_3_ showed enhanced light absorption in the visible and near-infrared region and remarkable photocatalytic performance [27]. However, some questions regarding the photocatalytic practical applications of SrTiO_3_ remain unanswered. The preparation conditions of these modified SrTiO_3_ photocatalyst are too complex, hazardous, and costly, thus hindering the extensive application of this compound in photocatalysis.

Supphasrirongjaroen et al. demonstrated that rapid quenching in different media can lead to the formation of Ti^3+^ self-dopants in TiO_2_, thus improving the photocatalytic activity of TiO_2_-based materials [28]. Herein, we report a facile method to prepare SrTiO_3_ rich in oxygen vacancies, with improved photocatalytic performance in the visible region. SrTiO_3_ was preheated to a high temperature, followed by rapid submersion in ethanol, resulting in the darkening of the color of SrTiO_3_. Various structural and electronic analyses revealed the introduction of oxygen vacancies in the quenched SrTiO_3_ (Q-SrTiO_3_), resulting in increased photocatalytic activity. The photocatalytic performance of Q-SrTiO_3_ was evaluated by using it for the degradation of rhodamine B (RhB) dye and the photolysis of water to produce H_2_, with the original SrTiO_3_ as a control. The results indicated the enhanced photocatalytic activity of Q-SrTiO_3_.

## 2. Materials and Methods

### 2.1. Preparation of Q-SrTiO_3_

1 g of original SrTiO_3_ nanoparticles (99.5%, ~100 nm, Macklin Biochemical, Germany) was weighed using an electronic scale, then 1 g of original SrTiO_3_ was put in a sintering boat (length 6 cm, width 3 cm, height 1.5 cm). When the Muffle furnace heat to 800 °C, the sintering boat along with 1 g of original SrTiO_3_ was transferred to the Muffle furnace. The SrTiO_3_ nanoparticles were heated at 800 °C for 20 min, and then open the Muffle furnace door, immediately took out the sintering boat and submerged the SrTiO_3_ in 45 mL ethanol (AR) at room temperature for rapid quenching. Afterwards, the quenched sample was filtered and then dried at 80 °C for 3 h for further use.

### 2.2. Characterization

An X-ray diffractometer (XRD, D8 ADVANCE, Bruker, Karlsruhe, Germany) was used to check the phase structures of all the samples. Data were collected between 20° and 80° (2θ) with a 0.02° step size using Cu Kα irradiation, at 36 kV tube voltage and 20 mA tube current. Field-emission transmission electron microscopy (FE-TEM, Talos F200S, FEI, Thermo, Hillsboro, OR, USA) and Field-emission scanning electron microscopy (FE-SEM, SU8220, Hitach, Tokyo, Japan) was used to determine the surface change, particle size and morphology of the samples. Sample specimens for the FE-TEM and the FE-SEM observations were prepared as follows: The powdered sample was dispersed in ethanol in an ultrasonic washing bath, and then, a drop of the suspension was dripped slightly onto a micro grid or a silicon slice and dried before imaging. X-ray Photoelectron Spectroscopy (XPS, Escalab 250Xi, Thermo Fisher, Bremen, Germany) was used to analyze the chemical composition and relative amount of the elements on the surface of the samples, with a reference of C1 s and the excitation source of 150 W Al Kα X-rays. An Electron paramagnetic resonance spectrometer (EPR, EMXplus-10/12, Bruker, Karlsruhe, Germany) was used to detect unpaired electrons in the samples at room temperature. A UV-Vis-NIR spectrophotometer (DRUV-vis, UV-3600 Plus, SHIMADZU, Kyoto, Japan) were used to record the UV-vis diffuse reflectance absorption in the range of 200–2000 nm.

### 2.3. Photocatalytic Test

#### 2.3.1. Photocatalytic Degradation of RhB

The photocatalytic degradation activities of Q-SrTiO_3_ were evaluated by monitoring the rate of decomposition of RhB in an aqueous solution under visible-light or UV-light irradiation from a 500 W Xe lamp equipped with a UV cut off filter (>420 nm) or 500 W Hg lamp, the light intensity of 500 W Xe lamp equipped with a UV cut off filter and 500 W Hg lamp are 6 mW/cm^2^ and 48 mW/cm^2^, respectively. A cylindrical Pyrex vessel equipped with a lamp was used as the photocatalytic reactor, with water circulation to keep the reaction temperature at about 27 °C. 40 mL of an aqueous solution of RhB (4 × 10^−5^ M) and 0.02 g Q-SrTiO_3_ or SrTiO_3_ were placed in a quartz tube for the degradation reaction. Before the photodegradation, a dark reaction was conducted for 30 min to ensure adsorption-desorption equilibrium between the photocatalyst and the RhB solution. Continuous magnetic stirring was carried out to keep the photocatalyst suspended in the RhB solution. Next, the mixture was exposed to visible light or UV light. The samples were collected at regular intervals (1.5 h or 1 h), and the concentration of RhB in the solution was determined using a UV-Vis spectrophotometer at 553 nm. The percentage of degradation was recorded as *C/C_0_,* and the reaction constant (K_app_) was calculated from the slope of the linear regression obtained from the plot of −ln (*C*/*C*_0_) vs. time, where C_0_ and C are the absorbance of the RhB solution initially and at a particular time, respectively.

#### 2.3.2. Photocatalytic Evolution of hydrogen

The photocatalytic H_2_ production experiments were conducted in a 400 mL Pyrex quartz glass reactor at normal pressure and temperature. The photocatalyst (100 mg) was dispersed in 100 mL of 10% aqueous methanol solution (methanol acting as a sacrificial agent) using a magnetic stirrer. Then the reaction mixture was dispersed in an ultrasonic washing bath for 10 min. Before the irradiation by a 300 W Xe lamp (CRL-HXF300, China) as the sunlight source, the reactor was deaerated with nitrogen gas. During the photocatalytic reaction, the reactant solution was maintained at room temperature by using a Low-temperature cooling circulating pump (CEL-CR300, China), and magnetic stirring was continually maintained to keep the photocatalyst suspended in the aqueous methanol solution. The amount of H_2_ generated was tested using an online Shimadzu GC-2014C gas chromatograph (Shimadzu, Japan) equipped with an MS-5A column. The total reaction time for each sample was 5 h, and the H_2_ concentration was measured every hour.

## 3. Results and Discussion

### 3.1. Characterization of the Photocatalysts

The peaks in the powder XRD patterns of SrTiO_3_ and Q-SrTiO_3_ (Figure 1) matched with the (100), (110), (111), (200), (210), (211), (220) and (310) planes, indicating a characteristic SrTiO_3_ cubic structure (JCPDS card: 73-6001). Moreover, a small peak for SrCO_3_ was observed, probably due to the coexistence of SrTiO_3_ and SrCO_3_ under the atmospheric operating conditions adopted in the hydrothermal method [29]. No other diffraction peak was observed in the XRD patterns. Comparison with the XRD patterns from the local enlargement of the diffraction peaks (inset of Figure 1) revealed a slight shift (2θ~0.15°) of the (110) peak to a lower angle for Q-SrTiO_3_. According to the Bragg equation (2d sin θ = λ. where d, θ and λ are the crystal spacing, diffraction angle, and X-ray wavelength, respectively), a shift in the diffraction peaks toward a lower angle suggests an increase in the lattice parameters. This might be attributed to the substitution of Sr^2+^ (ionic radius Sr^2+^ > Ti^4+^) for Ti^4+^ in Q-SrTiO_3_ [30].

The UV-visible absorption spectra of SrTiO_3_ and Q-SrTiO_3_ (Figure 2a) exhibited an absorption onset at ~400 nm, which corresponds to a band gap of 3.1 eV. In contrast to the SrTiO_3_, the photoabsorption of Q-SrTiO_3_ was dramatically enhanced in the both UV, visible and infrared light regions, consistent with the color change of the sample from white to gray (inset of Figure 2a). The improved light absorption was attributed to the formation of surface oxygen vacancies in Q-SrTiO_3_. Similar results have been observed in other studies [19,31].

EPR is highly sensitive to unpaired electrons; hence, it was used for the detection of oxygen vacancies and Ti^3+^ species in SrTiO_3_ and Q-SrTiO_3_ (Figure 2b) Both SrTiO_3_ and Q-SrTiO_3_ showed a distinct EPR signal at g = 1.977 and g = 2.002, which could be ascribed to Ti^3+^ and oxygen vacancies, respectively. Because of its intrinsic non-stoichiometry, SrTiO_3_ always contains a fraction of oxygen vacancies and Ti^3+^ ions [32]. EPR spectra revealed that Q-SrTiO_3_ exhibited a stronger signal intensity for oxygen vacancies than did SrTiO_3_, indicating the presence of more oxygen vacancies in Q-SrTiO_3_, thus favoring enhanced photocatalytic activity of the Q-SrTiO_3_. Takata and Domen also demonstrated that doping of a cation with a lower valence ion than that of the parent cation (such as Sr^2+^ in SrTiO_3_) can introduce oxygen vacancies, thus effectively improving its photocatalytic activity [32]. Thus, based on the XRD results, due to the exists of abundant oxygen vacancies, in order to keep the regional charge balance of the Q-SrTiO_3_, the Sr^2+^ ions could substituted for Ti^4+^ ions, so that lattice expansion occurs (Figure 2c) [32,33].

The high-resolution transmission electron microscopy (HR-TEM) images (Figure 3a,b) revealed the interplanar spacing of SrTiO_3_ and Q-SrTiO_3_ crystals to be ~0.27 nm, which is consistent with the d-spacings of the (110) crystallographic planes of cubic SrTiO_3_. However, Liu et al. have reported that an ice-water quenching TiO_2_ had introduced a disordered surface layer surrounding the crystalline core, and the surface lattice distortion is related to the generation of oxygen vacancies during the ice-water quenching [34]. In contrast, Q-SrTiO_3_ prepared in this work used 800 °C ethanol quenching that did not lead to specific disordered surface layer, therefore, the generation of more oxygen vacancies in Q-SrTiO_3_ have not introduced disordered surface layer. The SEM images (Figure 3c,d) revealed the particle size and particle morphology of Q-SrTiO_3_ and SrTiO_3_ show no change, and the average diameters of Q-SrTiO_3_ and SrTiO_3_ nanocrystals are ~100 nm, hence, ethanol-quenching can not change the particle size and particle morphology of samples.

XPS was used to investigate the surface chemical composition and VB position of SrTiO_3_ and Q-SrTiO_3_. The Sr 3d_5/2_, Sr 3d_3/2_, Ti 2p_1/2_, and Ti 2p_3/2_ binding energies were 133.2, 134.6, 458.0, and 464.0 eV, respectively, in accordance with the literature values (Figure 4a,b) [35]. The Sr 3d and Ti 2p spectra showed no obvious variation between SrTiO_3_ and Q-SrTiO_3_. The Sr/Ti ratio of the samples was estimated according to the peak area and sensitivity factor of Sr 3d and Ti 2p (Table 1). The atomic ratio of Sr to Ti on the surface was about 1.64 and 2.15 for SrTiO_3_ and Q-SrTiO_3_, respectively. The larger atomic ratio of Sr/Ti in Q-SrTiO_3_ than that in SrTiO_3_ might be due to the substitution of Sr^2+^ for Ti^4+^ on the surface of the former.

The O 1s high-resolution X-ray photoelectron spectra of SrTiO_3_ and Q-SrTiO_3_ showed two typical components of SrTiO_3_ (Figure 4c). The two peaks located at 529.2 and 531.5 eV were assigned to bulk oxygen and surface oxygen in the samples, respectively. Based on previous research, it was assumed that the peak intensity of surface oxygen was related to the concentration of oxygen vacancies on the surface of SrTiO_3_ and Q-SrTiO_3_ [19,36]. The peak intensity at 531.5 eV become stronger for Q-SrTiO_3_, indicating that the concentration of oxygen vacancies on the surface of Q-SrTiO_3_ increased after the ethanol-quenching process. The introduction of more oxygen vacancies on the surface of Q-SrTiO_3_ lattice resulted in impure/defect states in the band gap, enhancing the visible and near infrared-light absorption of Q-SrTiO_3_ [27]. Furthermore, the increased amount of oxygen vacancies can improve the efficient charge transport in Q-SrTiO_3_, followed by the Fermi level shift toward to the CB of the Q-SrTiO_3_, facilitating the separation of photogenerated electron-hole pairs and resulting in enhanced photocatalytic activity [19,27]. In the VB XPS profile, the VB maxima were estimated by linear extrapolation of the peaks to the baselines (Figure 4d). Both, SrTiO_3_ and Q-SrTiO_3_ displayed identical VB band positions at 2.3 eV below the Fermi energy, indicating no shift in the VB edge.

### 3.2. Photocatalytic Activity

The photodegradation of RhB in aqueous solution under visible-light irradiation was used to evaluate the photocatalytic activity of the Q-SrTiO_3_, with SrTiO_3_ as the control (Figure 5a). After adsorption-desorption equilibrium between the photocatalyst and the RhB solution was achieved in the absence of light, even a slight adsorption of RhB over the samples resulted in a slight decrease in the concentration of RhB. During the photodegradation process, SrTiO_3_ showed no appreciable reduction in the RhB concentration in aqueous solution; however, Q-SrTiO_3_ showed higher photocatalytic activity than SrTiO_3_ in reducing the concentration of RhB in aqueous solution. After 370 min of visible-light irradiation in the presence of Q-SrTiO_3_, RhB was decomposed by about 20%; in contrast, SrTiO_3_ caused only 3% decomposition of the dye. The reaction constant (K_app_) was calculated from the slope of the linear regression obtained from the plot of −ln (*C*/*C*_0_) vs. time (Figure 5b). These results suggested that Q-SrTiO_3_ shows better activity than SrTiO_3_ for the photodegradation of RhB under visible light.

The UV-light photocatalytic activities of Q-SrTiO_3_ were investigated by monitoring the decomposition of RhB in an aqueous solution, with SrTiO_3_ as the control (Figure 5c,d). After 270 min of UV-light irradiation, the RhB dye was almost completely decomposed (~90%) by Q-SrTiO_3_, (Figure 5c). The reaction constant (K_app_) was shown in Figure 5d, this results also suggested that Q-SrTiO_3_ shows better photocatalytic activity than SrTiO_3_ for the photodegradation of RhB.

The photocatalytic activity of Q-SrTiO_3_ for the photolysis of water to produce H_2_ in 100 mL 10% aqueous methanol solution was also studied under solar irradiation, using SrTiO_3_ as the control. Figure 5c,d present the time course of H_2_ generation for SrTiO_3_ and Q-SrTiO_3_ under solar light irradiation. Q-SrTiO_3_ steadily produced H_2_ gas at the rate of 42.12 μmol g^−1^ h^−1^, which was almost 6.2 times higher than that observe with SrTiO_3_ (6.83 μmol g^−1^ h^−1^). All these results demonstrated that Q-SrTiO_3_ possesses higher photocatalytic activity than SrTiO_3_.

## 4. Conclusions

In this paper, a facile and general method has been introduced to modify the surface environment of SrTiO_3_ through an ethanol-quenching process. Q-SrTiO_3_ showed higher photocatalytic activity than did SrTiO_3_ for the degradation of RhB and the photolysis of water to produce H_2_ under the irradiation by visible, UV or solar light. Results of spectroscopic characterization revealed that after rapid ethanol quenching, a high concentration of oxygen vacancies was introduced on the surface of the Q-SrTiO_3_ lattice. Consequently, in order to maintain the regional charge balance in Q-SrTiO_3_, the redundant Sr^2+^ is likely to substitute for Ti^4+^. Moreover, oxygen vacancies play an important role in enhancing the photocatalytic performance of Q-SrTiO_3_ by not only inducing localized states into the band gap of Q-SrTiO_3_, but also acting as photoinduced charge traps. Consequently, the light absorption ability is increased and the recombination rate of photogenerated electron-hole pairs is decreased, thus enhancing the photocatalytic activity of Q-SrTiO_3_.

## Figures and Tables

**Figure 1 nanomaterials-09-00883-f001:**
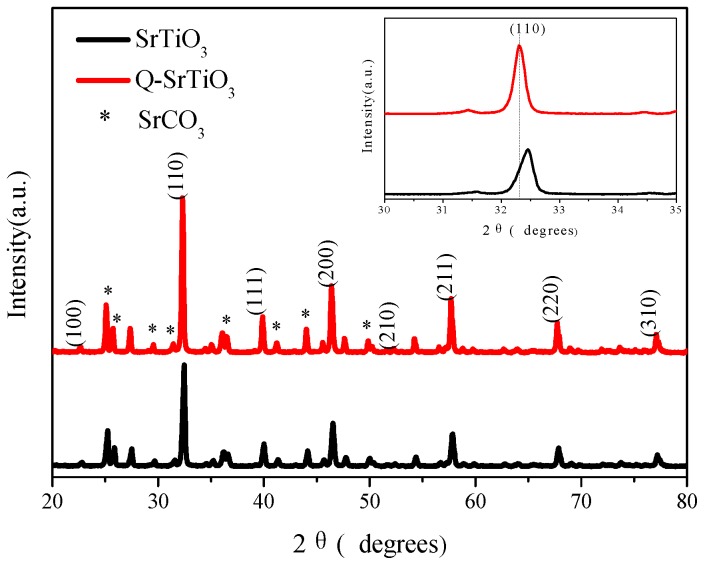
XRD patterns of SrTiO_3_ and Q-SrTiO_3_ samples.

**Figure 2 nanomaterials-09-00883-f002:**
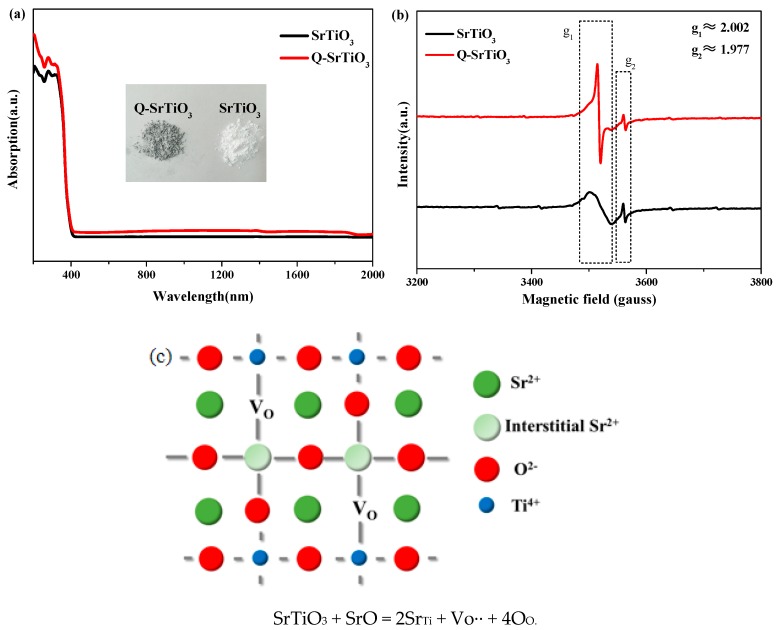
UV-vis absorption spectra of SrTiO_3_ and Q-SrTiO_3_ (**a**), the insert is a photograph of SrTiO_3_ and Q-SrTiO_3_; EPR spectra of SrTiO_3_ and Q-SrTiO_3_ (**b**). Schematic illustration of lattice change of Q-SrTiO_3_ after solvent-quenching (**c**).

**Figure 3 nanomaterials-09-00883-f003:**
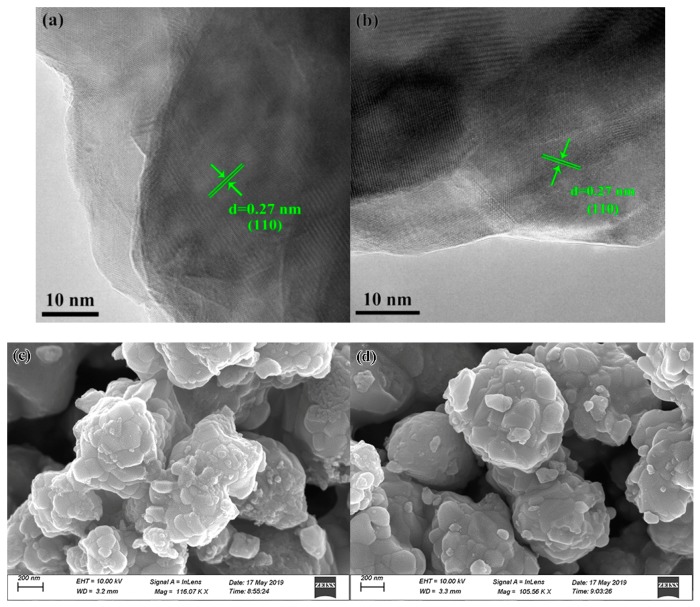
High-resolution TEM images and SEM images of SrTiO_3_ (**a**,**c**) and Q-SrTiO_3_ (**b**,**d**).

**Figure 4 nanomaterials-09-00883-f004:**
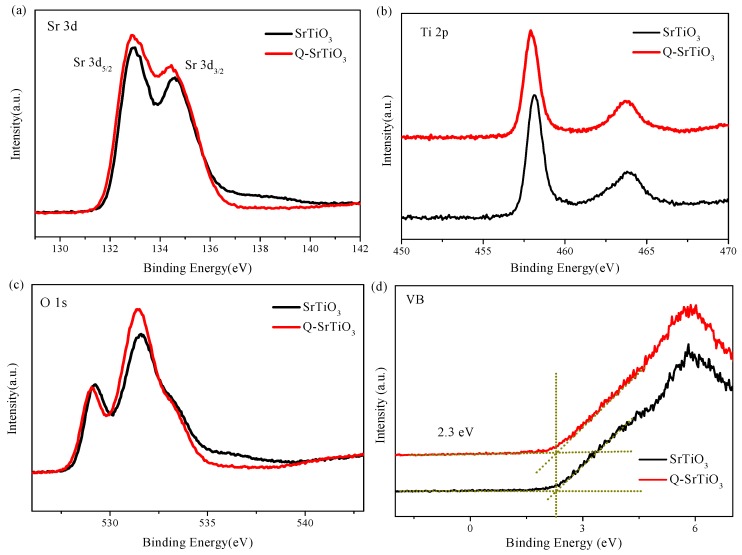
Sr 3d XPS spectra (**a**), Ti 2p XPS spectra (**b**), O1s Spectra (**c**), XPS valence band spectra of SrTiO_3_ and Q-SrTiO_3_ samples (**d**).

**Figure 5 nanomaterials-09-00883-f005:**
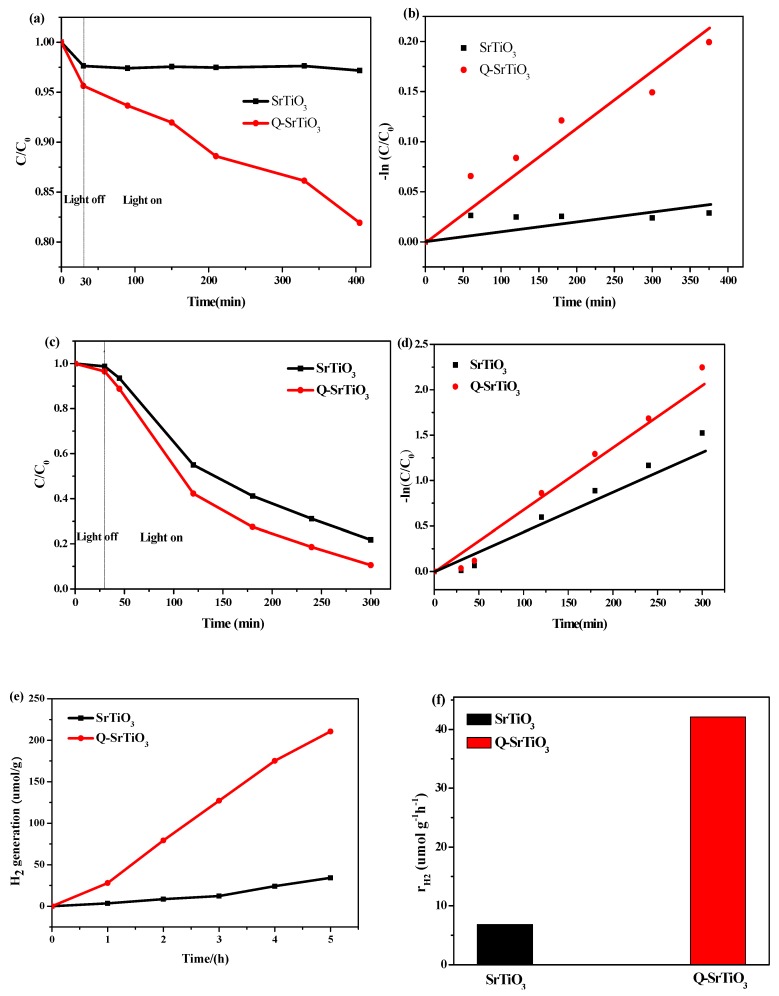
Photodecomposition of RhB aqueous solution with SrTiO_3_ and Q-SrTiO_3_ under visible light (**a**) and UV light (**c**). −ln (*C*/*C*_0_) of the RhB concentration as a function of visible light (**b**) or UV light (**d**) irradiation time. Time-course of photocatalytic water splitting for H_2_ generation in 100 mL 10% aqueous methanol solution under solar light (**e**). The rate of hydrogen generation for SrTiO_3_ and Q-SrTiO_3_ under solar light (**f**).

**Table 1 nanomaterials-09-00883-t001:** Summary of the XPS data for the SrTiO_3_ and Q-SrTiO_3_.

Sample	Atomic Concentration (%)				Atomic Ratio
	Ti	Sr	O	C	Sr/Ti
SrTiO_3_	3.73	6.14	31.36	58.76	1.64
Q-SrTiO_3_	2.83	6.10	29.75	61.32	2.15

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
