# Peer review of "Ethanol-Quenching Introduced Oxygen Vacancies in Strontium Titanate Surface and the Enhanced Photocatalytic Activity"

_nanomaterials, 2019, doi:10.3390/nano9060883_

Reviewer 1 Report

The paper is very interesting. It can be published after some revisions:

1) the authors should better describe the rapid quench with ethanol. I retain that this is a key aspect since the photocatalytic activity is induced by the oxygen vacancies and Ti3+ whose formation is promoted by this quenching step.

2) To check the visible light activity, the authors should also perform a test with a colourless pollutant (such as phenol).

3) I suggest to perform also photocatalytic tests under UV light to check if the sample modification (such as oxygen vacancies) may induce a higher photoactivity also under UV light.

Author Response

Reply to the Comments of Referee #1

Comment #1:

The authors should better describe the rapid quench with ethanol. I retain that this is a key aspect since the photocatalytic activity is induced by the oxygen vacancies and Ti3+ whose formation is promoted by this quenching step.

Response: Thank you for reminding us of the unclear writing about the synthetic method of the rapid quench with ethanol. We have added and rephrased as" then 1g of original SrTiO3 was put in a sintering boat (length 6 cm, width 3 cm, height 1.5 cm). When the Muffle furnace heat to 800 ℃, the sintering boat along with 1g of original SrTiO3 was transferred to the Muffle furnace. The SrTiO3 nanoparticles were preheated at 800 ℃ for 20 min, and immediately submerged in 45 ml ethanol (AR) for rapid quenching." according to the referee’s comment. See also text in lines 91-94 of page 3 in the revised manuscript.

Comment #2:

To check the visible light activity, the authors should also perform a test with a colourless pollutant (such as phenol).

Response:

Thank you for your suggestion. We agree that it is better to check the photocatalytic activity of conducting a test with a colourless pollutant under visible light, however, due to the limitations of our laboratory conditions, it is rather difficult for us to conduct an experiment of degrading colorless pollutants.

Comment #3:

I suggest to perform also photocatalytic tests under UV light to check if the sample modification (such as oxygen vacancies) may induce a higher photoactivity also under UV light.

Response:

Thank you for your suggestion. Additional experiments have been carried out to investigate the photocatalytic activity for the quenching SrTiO3 under UV light. Experimental data about photocatalytic degradation for the Q-SrTiO3 under UV light have been newly added as Figure 5(c) and Figure 5(d), and we also have made a detailed discussion in the revised manuscript. See also text in lines 246-250 of page 8 in the revised manuscript.

The newly added experimental data are attached below:

Figure 5. Photodecomposition of RhB aqueous solution with SrTiO3 and Q-SrTiO3 UV light (c). -ln (C/C0) of the RhB concentration as a function of UV light (d) irradiation time.

Reviewer 2 Report

The paper of Y. Wang et al on enhanced photocatalytic activity of strontium titanate lack of results to be accepted in nanomaterials. The photocatalytic test must be improve before resubmission.

1. It is claim that the photocatalytic activity was studied under solar irradiation (line 233). In fact a 300 W Xe lamp was used (500 W for RhB). Furthermore, the relative fraction of UV radiation in the emission spectrum of the Xe lamp is close to 50%. It would be nice to use filter to show that water splitting is/ is not possible under visible light using this new strontium titanate photocatalyst.

2. Furthermore, line 45, it is claim that "SrTiO3 could only be utilized only in the UV region of sunlight which largely restricts its pratical application". This point should be discussed further (see point 1 about emission spectrum of the Xe lamp).

3. No information of the intensity of irradiation was given

4. No information is given on the temperature and how the temperature is control (section 2.3.2) which is important working with 300W lamp for 5h.

5. 10% aqueous methanol solution is used (line 130) but this important point could not be found in the section 3.2 and in the caption of figure 5. The effect of MeOH concentration and reaction temperature should be, at least, studied.

As such (see point 1 to 5) discussion about photocatalytic activity (section 3.2) must be improve before resubmission.

6. The rate of photocatalytic hydrogen evolution is rather small (0.04 mmol h−1 g−1) whereas rate of 3.2 mmol h−1 g−1 could be obtained with SrTiO3 by the polymerized complex method under UV illumination (see Journal of Power Sources, 183 (2008) 701-707). Authors should comment.

7. security. The flash point of ethanol is 14°C. How is manage the security of the quench (SrTiO3 at 800°C immediately submerged in EtOH?) see line 92.

8. The results obtained with RhB should be removed or should be improved like for water splitting, especially adsorption study in the dark, analysis of the by-products ...

Author Response

Reply to the Comments of Referee #2

Comment #1

It is claim that the photocatalytic activity was studied under solar irradiation (line 233). In fact a 300 W Xe lamp was used (500 W for RhB). Furthermore, the relative fraction of UV radiation in the emission spectrum of the Xe lamp is close to 50%. It would be nice to use filter to show that water splitting is/ is not possible under visible light using this new strontium titanate photocatalyst.

Response: Thank you for this valuable suggestion. The light source of the 300 W Xe lamp (CEL-HXF300) that we used in the photocatalytic H2 production experiments is full-band light, and the wavelength is continuous distribution, therefore, it can be used to simulate sunlight that focuses on visible light. Before our first submission of this manuscript, we have conducted a photocatalytic H2 production experiment under visible light irradiation from a 300 W Xe lamp (CRL-HXF300) equipped with an UV-IRCUT 420 filter, the rate of photocatalytic hydrogen evolution of Q-SrTiO3 under visible light is very small, therefore, we will not discuss photocatalytic hydrogen production performance of the Q-SrTiO3 in visible light.

Comment #2

Furthermore, line 45, it is claim that "SrTiO3 could only be utilized only in the UV region of sunlight which largely restricts its pratical application". This point should be discussed further (see point 1 about emission spectrum of the Xe lamp).

Response: Thank you for your suggestion. According to other literature (J. Phys. Chem. B., 2004, 108, 8992-8995, and Appl. Catal. B., 2016, 186, 97-126.), SrTiO3 is a wide band gap photocatalyst (3.1-3.7 eV). Therefore, on the basis of the formula: Eg=1240/λ, the corresponding wavelength of absorption can be calculated as 335-400 nm, SrTiO3 could only conduct a photocatalytic reaction under ultraviolet light irradiation.

Comment #3

No information of the intensity of irradiation was given

Response: Sorry for our not mentioned description of irradiation intensity of the lamp. We have test the irradiation intensity of 500W Xe lamp and 500W Hg lamp, and the irradiation intensity are about 6 mW/cm2 and 48 mW/cm2, respectively. We have added this data in the revised manuscript, see also text in lines 120-122 of page 3 in the revised manuscript. However, based on the condition of present work, it is rather difficult for us to test the irradiation intensity of the 300 W Xe lamp.

Comment #4

No information is given on the temperature and how the temperature is control (section 2.3.2) which is important working with 300W lamp for 5h.

Response: Thank you for reminding us of the not mentioned writing about that: “ the temperature and how to control the temperature during the working with 300W Xe lamp for 5h”. A built-in fan was used to control the temperature of the 300 W Xe lamp, a Low-temperature cooling circulating pump was used to maintained the temperature of the reactant solution at room temperature, and an external fan assist in cooling the whole system. We have added the details according to the referee’s comment of how to control the temperature of the reactant solution and 300W lamp during the reaction, see also text in lines 140-141 of page 4 in the revised manuscript.

Comment #5

10% aqueous methanol solution is used (line 130) but this important point could not be found in the section 3.2 and in the caption of figure 5. The effect of MeOH concentration and reaction temperature should be, at least, studied.

Response: Thank you for your suggestion. I'm sorry we missed the important point that 10% methanol solution is used as a sacrificial agent in photocatalytic hydrogen production. For rigorous writing, the sentence" in 100 ml 10% aqueous methanol solution" has been added in the section 3.2 and the caption of Figure 5. See also the text in lines 251-252, 262-263 of page 9, 10 in the revised manuscript. The reactant solution was maintained at room temperature by using a Low-temperature cooling circulating pump. Cryogenic cooling is used to cool reaction solution, in case the reaction temperature is too high that make photocatalyst and sacrificial agent thermolabile, or the reaction rate is too high that the experimental data were inaccurate. Therefore, we have controlled the effect of reaction temperature on catalyst and sacrificial agent as far as possible. The concentration of methanol solution was selected according to other literature (J. Phys. Chem. B 2004, 108, 8992-8995), however, because of the limited experimental conditions, it is rather difficult for us to study the effect of aqueous methanol concentration on photocatalytic hydrogen production.

Comment #6

The rate of photocatalytic hydrogen evolution is rather small (0.04 mmol h−1 g−1) whereas rate of 3.2 mmol h−1 g−1 could be obtained with SrTiO3 by the polymerized complex method under UV illumination (see Journal of Power Sources, 183 (2008) 701-707). Authors should comment.

Response: Thank you for your suggestion. We have read this article carefully (Journal of Power Sources, 183 (2008) 701-707 ). According to the conclusion in this articles, the particle size affect the photocatalytic activity of SrTiO3 for hydrogen evolution, the average diameters of Q-SrTiO3 and untreated SrTiO3 are 100 nm (SEM image was attached below), however, the mean diameters of the as-synthesized SrTiO3 particles were 30 nm by the polymerized complex method, which possess a small size. Therefore, it should be favorable for photocatalytic hydrogen production. Besides, the surface of the SrTiO3 samples synthesized by the polymerized complex method was loaded cocatalyst Pt to make photocatalytic hydrogen evolve easily, and the irradiation source was UV light from a 500 W high pressure mercury lamp, however, no cocatalyst was used in our photocatalytic hydrogen production experiment, and the irradiation source was simulative solar light from a 300 W Xe lamp, as a result, the rate of photocatalytic hydrogen evolution of quenching SrTiO3 is far less than they do. In addition, we just compared the rate of photocatalytic hydrogen evolution of Q-SrTiO3 with untreated SrTiO3.

Figure 5(c),(d). SEM images of untreated SrTiO3 and Q-SrTiO3.

Comment #7

security. The flash point of ethanol is 14°C. How is manage the security of the quench (SrTiO3 at 800°C immediately submerged in EtOH?) see line 92.

Response: Thank you for your suggestion, we have carried out many quenching tests , it is very safe during SrTiO3 rapidly submerged in ethanol. When SrTiO3 is heated to a higher temperature and immediately submerged in ethanol, the ethanol vaporization and be on fire, therefore, an iron pan was used to cover the burning ethanol.

Comment #8

The results obtained with RhB should be removed or should be improved like for water splitting, especially adsorption study in the dark, analysis of the by-products.

Response: Many thanks for pointing out this important issue. We agree that the adsorption study in the dark and analysis of the by-products could better study the photocatalytic degradation process, however, based on current experimental conditions, it is rather difficult for us to conduct a adsorption study in the dark and analysis of the by-products.

Reviewer 3 Report

This manuscript reports the ethanol quenching of SrTiO3 crystals and the resulting photocatalytic properties of the starting ceramic and the quenched material. Materials were characterized by various techniques including XRD, EPR, HRTEM; photocatalytic testings demonstrate that quenched material has far enhanced H2-generation activity under UV-vis irradiation over the starting SrTiO3 due to interstitial Ti+4 substitution by Sr+2 in the quenched material. Method is simple and allow for rapid preparation of SrTiO3 photocatalyst materials. Results are of interest to a scientists and engineers in the fields of nanomaterials, energy storage and conversion, environmental chemistry and photocatalysis. I have only minor comments in order to improve the material. Once these are addressed, no further revisions are needed. My comments are:

Please calculate the band gap for materials using UV-vis.

Authors purchased nanosized SrTiO3 from a commercial supplier as starting material. What is the average particle size and particle morphology (if well-defined or ill-defined) for this material, and did ethanol quenching affect those properties? SEM or TEM may provide that information.

Author Response

Reply to the Comments of Referee #3

Comment #1

Please calculate the band gap for materials using UV-vis.

Response: Thank you for your suggestion. We have calculated the band gap of SrTiO3 using UV-vis, according to UV-vis spectra, an absorption onset at ~400 nm was observed, the band gap of quenching SrTiO3 and untreated SrTiO3 was calculate with the formula: Eg=1240/λ nm. We have rephrased the sentence as” The UV-visible absorption spectra of SrTiO3 and Q-SrTiO3 (Figure 2 (a)) exhibited an absorption onset at ~400 nm, which corresponds to a band gap of 3.1eV.” in the revised manuscript, see also the text in lines 162-163 of page 5 in the revised manuscript.

Comment #2

Authors purchased nanosized SrTiO3 from a commercial supplier as starting material. What is the average particle size and particle morphology (if well-defined or ill-defined) for this material, and did ethanol quenching affect those properties? SEM or TEM may provide that information.

Response: Thank you for this valuable suggestion. We have taken many TEM and SEM photographs (the SEM photographs are shown attached below), and we have added the information of Field-emission scanning electron microscopy in section 2.2. After ethanol quenching process, there was no significant change in average particle size and particle morphology. We have added the sentence as" The SEM images (Figures 3(c),3(d)) revealed the particle size and particle morphology of SrTiO3 and Q-SrTiO3 show no change, and the average diameters of Q-SrTiO3 and SrTiO3 nanocrystals are ∼100 nm, hence, ethanol-quenching can not change the particle size and particle morphology of samples.", see also the text in lines 194-197 of page 6 in the revised manuscript.

Figure 3. SEM images of the SrTiO3 (c) and Q-SrTiO3 (d).

Round  2

Reviewer 1 Report

The authors clarified all the aspects underlined by me.

Author Response

Reply to the Comments of Referee #1

Comment #1:

The authors clarified all the aspects underlined by me.

Response: We greatly appreciate the referee for his/her recognition on our revised manuscript and the efforts to improve the quality of our manuscript.

Reviewer 2 Report

The authors correct their paper according to my comments (UV, MeOH, ...) and thus it could be published. I personally do not think that the paragraph with the decomposition of RhB in an aqueous solution was innovative and interesting.

Safety issue for the quenching part (SrTiO3 nanoparticles at 800 ℃ immediately submerged in 45 ml ethanol) must be improved and a paragraph with the safety issues of the procedure should be highlights: working in hood, fire blanket ?, dry chemical fire extinguisher ? ….

Author Response

Reply to the Comments of Referee #2

Comment #1:

The authors correct their paper according to my comments (UV, MeOH, ...) and thus it could be published. I personally do not think that the paragraph with the decomposition of RhB in an aqueous solution was innovative and interesting.

Response: Thank you for your recognition on our revised manuscript and the efforts to improve the quality of our manuscript. Many literatures have evaluated the photocatalytic properties of samples by photodegradation of RhB in an aqueous solution in recent year (Sci. Rep., 2016, 6, 38064, Appl. Catal., B, 2015, 176, 654-666 and Appl. Surf. Sci., 2017, 391, 360-368.), therefore, we conducted photodegradation of RhB to evaluate the photocatalytic activity of samples.

Comment #2:

Safety issue for the quenching part (SrTiO3 nanoparticles at 800 ℃ immediately submerged in 45 ml ethanol) must be improved and a paragraph with the safety issues of the procedure should be highlights: working in hood, fire blanket ?, dry chemical fire extinguisher ? …

Response: Thank you for this valuable suggestion. It is very safe during SrTiO3 rapidly submerged in ethanol, in most cases, after the quenching process, nothing dangerous happened, even if the ethanol be on fire (the fire are low flame), we also have corresponding security measures to solve, for example, a fire blanket or iron pan was used to cover the burning ethanol. We believe that the clear writing about the synthetic method of the rapid quench with ethanol should be rephrased. See also text in lines 93-96 of page 3 in the revised manuscript.

Reviewer 3 Report

This work reports the ethanol quenching of SrTiO3 crystals and the resulting photocatalytic properties of the starting ceramic and the quenched material. This work is of interest to scientists and engineers working on materials design and applications to environmental sciences, catalysis, separations, and energy conversion. The manuscript has been sufficiently revised following reviewers' comments. It is now recommended for publication.